# An Assessment of Mental Health Challenges of Lesbian, Gay, Bisexual, and Transgender College Students during the COVID-19 Pandemic

**DOI:** 10.3390/healthcare12202047

**Published:** 2024-10-16

**Authors:** Lei Xu, H. Daniel Xu, Wenhua Lu, Divya Talwar

**Affiliations:** 1Department of Health Education and Promotion, College of Health and Human Performance, East Carolina University, 300 Curry Court, Greenville, NC 27858, USA; 2Department of Political Science, East Carolina University, Greenville, NC 27858, USA; xuh19@ecu.edu; 3Department of Community Health and Social Medicine, School of Medicine, The City University of New York, New York, NY 10031, USA; wlu1@med.cuny.edu; 4Children’s Hospital of Philadelphia, Philadelphia, PA 19104, USA; talward@chop.edu

**Keywords:** mental health, college students, LGBTQ+ group, emergency management, COVID-19

## Abstract

Introduction: Collegiate mental health continues to be a worrisome public health concern among college students in the U.S. The unprecedented COVID-19 pandemic has caused an upward trend of mental health crises, especially among lesbian, gay, bisexual, and transgender (LGBTQ+) college students. The objective of this study was to assess the mental health statuses, attitudes towards disease control and mitigation measures, and coping strategies among this vulnerable group. Methods: A web-based survey was conducted at a medium-sized public university in the mid-Atlantic region during the summer and fall of 2021 when COVID-19 was still a major public health concern. The survey was distributed through the listservs of the college and was advertised through campus-wide social media. Descriptive and inferential statistics including a t-test for the differences in group means and a logit regression model for comparing the groups were used. Results: Our final sample is composed of 611 students with 79% of the respondents identifying as straight, and 20% in the LGBTQ+ group. Our results showed that LGBTQ+ students exhibited higher levels of anxiety and fear compared to the non-LGBTQ+ groups. Also, a large proportion of LGBTQ+ students were negatively impacted by the pandemic as compared to the non-LGBTQ+ groups (*p* = 0.05), while they generally have more positive views on the public health measures to alleviate the adverse impacts from COVID-19 (*p* = 0.001). Conclusions: Public health emergency management should adopt appropriate strategies and adapt their services to support the mental health needs of LGBTQ+ students. Our study highlighted the need to design tailored health promotion programs and enhance support systems for LGBTQ+ college students during similar emergencies.

## 1. Introduction

There has been an increased focus on assessing the impacts of emergencies and disasters on minority and vulnerable populations [1]. The unprecedented COVID-19 pandemic emergency has disproportionally impacted these populations as many of them are socially and economic disadvantaged and may be less prepared for natural disasters [2,3]. The mental health crises resulting from these emergencies remain a serious public health concern [4]. As such, vulnerable groups may need different coping strategies in response to their fear and stress in emergencies. Prior researchers have identified several key components to enhance the effectiveness of emergency management and have emphasized culturally competent care and language access services; organizational supports are important for disaster response partners, such as first responders, disaster response professionals, community members, and members of state and local governments [5,6,7,8]. However, limited practice and research of emergency management exist to serve lesbian, gay, bisexual, transgender, queer, and questioning (LGBTQ+) communities, especially LGBTQ+ college students [9,10,11].

Mental health disparities in LGBTQ+ college students have been extensively reported, and, not surprisingly, these studies highlighted the heightened levels of anxiety, depression, and mental distress among these groups [12,13,14,15,16,17]. Studies have called to attention that the situation may worsen among the LGBTQ+ students who sheltered at home with their families who were fearful to share with them their sexual identities during the COVID-19 pandemic [15,17]. LGBTQ+ students also showed high unmet medical and mental health needs and expressed concerns related to personal finances, worries about safety, and the difficulties to disclose their LGBTQ+ identities [11,14] As such, clear evidence demonstrated the urgent need to enhance mental health services for LGBTQ+ college students during the COVID-19 pandemic [10,11,14].

Although there has been an increased focus on diversity, equity, and inclusion in emergency management for higher education before the pandemic [18,19], more efforts could be set forth to include LGBTQ+ groups in these programs. Recent research of the pandemic preparedness after COVID-19 focuses on the medical students, dental students, and other health profession students but is yet to involve minority groups such as college LGBTQ+ groups [20,21]. This study attempts to enhance our knowledge about LGBTQ+ students’ mental health needs and challenges among southern college student populations one year into the COVID-19 pandemic.

## 2. Methods

### 2.1. Setting and Participants

A web-based survey was conducted at a public university in the mid-Atlantic region of the U.S. during the summer and fall of 2021 when COVID-19 was still a major public health concern. Registered undergraduate students who are over 18 years old were eligible to participate and they all provided online consent before participating.

### 2.2. Survey Design

This current study was part of a lager research project assessing college students’ mental health statuses. A team of researchers specialized in the field of social and behavioral science, psychology, and public health designed the current study based on an extensive review of existing literature on the impact of COVID-19 on mental health among college students [22,23,24,25,26,27,28,29,30,31]. The survey was reviewed and approved by the university’s Institutional Review Board under an exempt category prior to the beginning of the research [UMCIRB 20-002109]. The entire survey contained 80 items and eight sections, including (1) questions regarding participants’ demographic information, (2) perceptions of COVID-19 and the COVID-19 vaccine, (3) anxiety levels resulting from COVID-19 (GAD-7 anxiety scale), (4) levels of fear resulting from COVID-19 (self-rating fear of COVID-19 scale), (5) academic stressors from the effects of COVID-19, (6) coping strategies and behavioral changes during COVID-19, (7) support systems in place during COVID-19, and, lastly, (8) barriers regarding the utilization of mental health services. The items in the questionnaire included both questions with Likert scales and open-ended questions for the participants to elaborate on their answers. The details of the survey items were provided elsewhere in our previously published work [32].

### 2.3. Survey Distribution

The survey was distributed via Qualtrics^TM^ 2020 (an online software free for faculty and students at the host university) through the help of the university’s Institutional Planning, Assessment and Research (IPAR) office in the first stage. Then we utilized social media (Instagram and Facebook) as well as department and college email listservs to promote the visibility of this study and recruitment. The two sequential ways of survey distribution helped us reach 4666 students in total. A cover letter to recruit potential students and multiple follow-up reminders were also sent to the students. There were 611 valid responses with a 16.4% response rate.

### 2.4. Statistical Analyses

Descriptive and inferential statistics were used for the data collected for the purpose of our study. Specifically, descriptive statistics were used in analyzing demographic data, whereas inferential statistics including a *t*-test for differences in group means and a logit regression model for comparing the probability (odds ratio) of differences in various indicators between LGBTQ+ and non-LGBTQ+ students were applied. The level of significance for statistical tests applied in this study is 0.01, with an additional 0.05 indication for lower levels of significance in light of the exploratory nature of this study. STATA 16 [33] was used for the statistical analysis.

## 3. Results

### 3.1. Demographics of the Respondents

Our key study population is college LGBTQ+ students who were enrolled in various programs and of various race and age groups during the COVID-19 pandemic. Table 1 provides the demographic profile of the participants. Among 611 students, 20% self-identified as non-heterosexual, including asexual, bisexual, gay, lesbian, pansexual, queer, and unknown; 79% aged 17–23 years; 61% were white; and 72% were junior and senior students. In terms of students’ fields of study, around 64% of students were from the programs in health and health performance, arts and sciences, and business. About 27% of the respondents indicated they had been infected with COVID-19, and 52% had at least a family member diagnosed with COVID-19.

### 3.2. Mental Health Impact and Coping Strategies

We analyzed the data to see to what extent LGBTQ+ students were impacted by the pandemic in terms of their mental health. Overall, a large proportion of LGBTQ+ students were negatively impacted by the pandemic as compared to the non-LGBTQ+ groups (72% vs. 81%, *p* = 0.05). Specifically, their negative experiences stemmed from a broad range of issues (*p* < 0.001), such as the negative impact on daily life, mental health, and perceived anxiety/stress of their peers, and they generally had more positive views on the public health measures to alleviate such adverse impacts of COVID-19 (2.4 vs. 4.0, *p* < 0.001, as in Table 2).

Another important aspect the study investigated is to see if there were any significant differences in the coping strategies between LGBTQ+ and non-LGBTQ+ students during the pandemic. Overall, our data suggest that almost half of the students experienced an elevated level of fear for the ongoing pandemic (48%). However, the academic stress scale measured by a battery of seven questions is significantly higher among the LGBTQ+ group than other groups (*p*-value = 0.025). Some indicators for academic stressors include missed assignment deadlines, dropping of grades, difficulty focusing, and reduced interaction with classmates. However, it appears that LGBTQ+ students used a set of different strategies to cope with the mental health challenges, ranging from healthy strategies, such as learning a new hobby or skill and increased exercising, to unhealthy or harmful strategies, such as increased consumption of unhealthy foods, increased use of social media, and increased smoking of controlled substances. LGBTQ+ students seemed to use unhealthy strategies more (*p*-value < 0.05) and use healthy strategies less frequently than other groups (*p*-value = 0.06) (Table 3). LGBTQ+ groups experienced a higher level of anxiety and fear and faced more challenges in obtaining social support. Compared to non-LGBTQ+ students, they felt they were more distanced from their immediate and extended families and lacked a strong support system during the pandemic (*p* < 0.01).

### 3.3. Financial Impact and Attitude toward Public Health Measures

As LGBTQ+ college students are generally more economically vulnerable, this study also investigated the pandemic’s impact on their financial wellbeing, which may complicate their mental health. Approximately 44% of the respondents indicated that their financial status had somewhat or significantly worsened during the pandemic. Meanwhile, students who experienced financial decline also tended to experience increased anxiety and fear (OR = 2.5, *p* < 0.01; OR = 1.87, *p* < 0.01, respectively). In addition, while about 73% of the respondents overall reported an increase of anxiety, 81% of the LGBTQ+ respondents indicated they had experienced heightened anxiety and a higher percentage of LGBTQ+ students had experienced financial decline (47%). Similarly, while about 48% of the respondents reported an increase in fear, 60% of the LGBTQ+ students stated an increased fear with a relatively higher percentage of them experiencing worsened financial conditions.

Our data also suggest that LGBTQ+ groups were 1.4 times more willing to be vaccinated than non-LGBTQ+ groups (Table 3). Overall, about 53% of the respondents indicated that they are either certainly or likely to be vaccinated within three months. In contrast, 63% of LGBTQ+ respondents indicated that they would certainly or likely be vaccinated as compared to only 51% for other groups. This difference is understandable when it is compared to their different beliefs about the effectiveness of the vaccine (*p* < 0.01), Specifically, a significantly larger percent of straight students held negative views on the effectiveness of the vaccine (18% vs. 7%). In terms of the sources of information to learn about COVID-19, the respondents seemed to equally rely on a mix of government sources and health professionals (48%) and other sources, such as friends and family, news sources, and social media. The sources of information such as social media can be highly influential in student’s perceived risks and their views on the effectiveness of public health measures other than vaccines, such as masking and social distancing. Our analysis found that these public health measures acted as powerful predictors for students’ anxiety (OR = 0.10, *p* < 0.01) and fear (OR = 0.12, *p* < 0.01).

## 4. Discussion

The uniqueness of this current study was we retrospectively examined the mental health challenges among the college LGBTQ+ groups from a public health emergency preparedness perspective. The findings from this study highlighted the need to inform different LGBTQ+ health stakeholders when they design and implement health interventions and help them enhance the pandemic preparedness plans in college settings. The manifestations of LGBTQ+ students’ psychological distress ranged from mild to more severe, i.e., from feeling nervous, anxious, or on edge, worrying too much about how COVID-19 may affect their lives, to being so restless that it is hard to sit still. Based on our findings, several public health emergency management approaches should be considered when preparing for LGBTQ+ groups when facing significant outbreaks of infectious disease.

First and foremost, our study echoed other studies examining mental health manifestations during the COVID-19 pandemic among college students in the U.S. in general, showing that students had heightened mental health dilemmas, such as increased stress, anxiety, and depression during COVID-19 [22,23,24,25,34]. Most of the prior research explored students’ mental health statuses at the initial stage of COVID-19 and mentioned that students’ mental health statuses worsened primarily due to school closures and disrupted educational and personal lives [22,23,31,35]. For example, Wang et al. surveyed over 2000 U.S. college students in May 2020, and found that 38–48% reported they experienced moderate-to-severe anxiety and depression [31]. Furthermore, Roberts and colleagues concluded that their longitudinal data indicate the persistence of students’ mental health struggles across the 2020–2021 semesters [35]. However, the majority of these studies did not provide findings pertaining to the mental health statuses of vulnerable groups, such as LGBTQ+ groups who had experienced high levels of fear, stress or anxiety before the outbreak of the pandemic. Our study added novel evidence to the existing literature, reporting the worsened mental health among LGBTQ+ college students one year into the COVID-19 pandemic. Moreover, our analyses showed that LGBTQ+ college students continued to experience higher levels of overall anxiety and fear than their non-LGBTQ+ counterparts immediately after the students returned to campus after the closures due to the pandemic.

More specifically, our study suggested that a concerning number of the LGBTQ+ students’ lives were more negatively impacted emotionally and socially than non-LGBTQ+ group after they experienced the pandemic for a whole year. Alarmingly, they tended to seek more negative coping measures while living during the pandemic. The LGBTQ+ students in our sample expressed more favorable or supportive attitudes towards the public health measures taken by the university, which was in contrast with the heterosexuals or straight students. They were more likely to support the measures mandated by the university during the time the data were collected, such as the university should enforce effective social distancing procedures on campus and require the students to wear masks in enclosed buildings. This finding sent off a positive signal showing that colleges and universities could be the place to provide effective counseling services or interventions to sexual minority college students on how to better cope with their anxiety, fear, and other mental health challenges. A study conducted before the pandemic [21] suggested that LGBTQ+ students were more likely to seek and receive on-campus mental health services compared to non-LGBTQ+ students. In a more recent study conducted at the initial stage of the pandemic, Goznzales et al. [14] mentioned that about 90% of the LGBTQ+ students felt the threat of the pandemic, 40% of the LGBTQ+ students reported unmet mental health needs, and 28% were concerned about seeking care due to their LGBTQ+ identity. Student health centers and resource centers could play a paramount role in LGBTQ+ student health and have been reported to be the preferred places for disclosing health concerns among LGBTQ+ students [21,36]. College counseling services should conduct needs assessments among the LGBTQ+ students to truly understand what will benefit the students most if another unexpected pandemic occurs.

Our study provided more nuanced pictures and showed that LGBTQ+ juniors experienced higher levels of anxiety. It will be beneficial that college counseling services could consider diversifying the mental health programs and attend to the different needs between students from different years, majors, and departments. Although many colleges and universities addressed special attention to students’ mental health crises and recruited more mental health specialists for their campuses after the COVID-19 pandemic, the LGBTQ+ students might still feel reluctant to go to the student counseling centers if they feel the services are “fit-for-all” students and are not intended for their needs. In other words, knowing exactly what the unmet needs are could help mitigate the existing gaps associated with mental health services and involving representatives from the LGBTQ+ groups, LGBTQ+ resource centers, college and university administrators, and other related stakeholders in emergency preparedness training is warranted.

Secondly, our analysis identified public health measures, including mandatory distancing, mask wearing, and vaccinations, could act as a powerful predictor for students’ anxiety and fear during the COVID-19 pandemic. One of the key measures to execute the pandemic preparedness plan is to promote the vaccine uptake among the priority population [23]. LGBTQ+ students in our study expressed more supportive attitudes towards the effectiveness of the COVID-19 vaccine to cope with the pandemic than non-LGBTQ+ groups. We also found that our sample of LGBTQ+ students expressed more confidence and willingness to obtain the COVID-19 vaccine than their counterparts. This finding resonated with the recent study by the U.S. Centers for Disease Control and Prevention (CDC), revealing that gay and lesbian adults were more receptive towards the COVID-19 vaccine and were more likely to be vaccinated [35]. However, it was also pinpointed that gay men were more likely to be vaccinated and only 50% of African American lesbians obtained the vaccine. Based on the results from the literature reviews exploring the hesitation and factors of the COVID-19 vaccine refusal, concerns about vaccine safety, efficacy, and the lack of trust in medical professionals were all mentioned as common barriers [37,38,39,40,41]. Further studies need to investigate the subgroup nuances (such as the differences by race/ethnicity, by gender identity, and by regions) that could fully unveil health inequalities associated with vaccine hesitation or refusals within the college LGBTQ+ community, which could potentially help local public health agencies design or tailor education interventions among different subgroups within this community and ensure they are up to date with the recommended vaccinations.

Thirdly, it is imperative that colleges or universities recruit public health specialists who are equipped with the knowledge and skills to attend the psychological needs of the LGBTQ+ groups on campuses. Many colleges in the United States have been struggling to meet students’ increased demands for mental health services and complaints of the insufficiency of on-campus counseling and mental health specialists [42]. Public health specialists with special training on behavioral changes are in a better position to provide such education, support, and services. Besides adding more skilled public health professionals, peer counseling is another promising strategy that could potentially help LGBTQ+ students with their mental health crises [43]. Peer counseling provides a platform and allows trained LGBTQ+ students with shared identities and experiences to help their peers with mental health concerns. It can be provided during the times when they are unsure about the need to seek professional counseling, or when they simply need to talk. Furthermore, the long waiting times and scheduling issues could be other barriers that prevent LGBTQ+ students from seeking mental health counseling during their crises. Therefore, telehealth/teletherapy options could be one effective alternative method and should be utilized to enhance LGBTQ+ students’ mental health needs during the pandemic [30]. Digital platforms that can foster the care and understanding among the LGBTQ+ community during the pandemic need to be established. Privacy and confidentiality should be ensured before inviting the members from the LGBTQ+ groups to join the digital platforms.

Fourthly, our study also accentuates the need to enhance support systems during the pandemic emergency for the LGBTQ+ student groups. Although LGBTQ+ students in our study experienced a higher level of anxiety and fear, they faced more challenges in obtaining social support as compared to other groups. This finding echoed prior research showing LGBTQ+ groups appeared to experience a lack of support and felt more distant from their immediate and extended families and friends [44]. It is worth mentioning that a lack of support systems has been proven to exacerbate the LGBTQ+ groups’ mental health situation during the COVID-19 pandemic [45,46]. For students who chose to conceal their gender identity and sexual orientation from their families and friends due to fear or other emotional distress, they could experience higher levels of depression or mental distress [15,33]. Special care and attention need to be provided to them during a public health emergency due to unprecedented stresses these LGBTQ+ youth might need to face.

Fifthly, emergency planners and college administrators need to create a more healthy, nurturing, and supportive college campus environment (such as temporary shelters or a food bank) that allows LGBTQ+ students who express their anxiety and fear of having to be sent back home during a pandemic emergency [14]. Colleges should consider providing safe and accessible living places that can accommodate the LGBTQ+ groups on campus or near the campus. Many studies have reported young LGBTQ+ students who experienced higher anxiety were confined with their parents during the lockdown periods due to the public health emergency for COVID-19 [13,14,15,16,17]. A recent study has recommended that colleges should consider mitigating some dilemmas such as hunger, isolation, financial challenges, and lack of access to the internet and/or technology, which may impact all students but could be particularly experienced by LGBTQ+ students during the COVID-19 pandemic. More studies should identify some options that could be provided to the students in such situations as returning to college campuses, what emergency housing plans can be established, and what gender-inclusive policies should be implemented for improving LGBTQ+ students’ mental health wellbeing.

Lastly, we found that LGBTQ+ groups in our study mainly rely on the information and resources from a mix of government sources and health professionals. Health programs with educational components that could potentially help the students with increased mental health stresses should be promoted at the county or regional levels. Several large-scale mental health promotion programs targeted at college students due to COVID-19 have been implemented in U.S. colleges or universities. For instance, North Carolina Governor Roy Cooper directed USD 5 million from the Governor’s Emergency Education Relief fund to implement mental health initiatives across the state’s colleges and universities, such as Mental Health First Aids (with both online and hybrid modes) to address mental health needs among college students in North Carolina [47]. Such fundings enabled higher education institutions to make immediate investments in services, staff, and programming to help more students. However, since these programs were designed before the pandemic, they could hardly keep up with the rapid changes of the mental health conditions caused by the pandemic and, due to the lack of evaluation plans, it was difficult to determine the effectiveness of such programs. In addition, such programs were provided to the college students in general, so LGBTQ+ students might not find them suitable to them. Again, some issues with such programs are usually constrained by the short duration, a lack of participation incentives, and a lack of sustainability. Understanding students’ mental health struggles and addressing their mental health concerns with workable programs should be a consistent pursuit instead of only implementing such programs during or after a crisis emerges and lasting only for a limited period of time. State-level policies that address LGBTQ+ mental health issues should be institutionalized for the best benefits by supporting the psychosocial wellbeing of the LGBTQ+ groups.

### Limitations

This study is not without limitations. One limitation is the bias in sampling. The sample may not be representative of the colleges and universities in the U.S. For instance, the students who had been more affected by the COVID-19 pandemic may have been more likely to respond to the online survey and, therefore, are overrepresented in the sample. Further, the survey was administered online, which may have skewed the sample as some students may not have adequate online access to respond to the survey. Future research should include colleges of various sizes and types, with different geographical and cultural environments. However, the findings from our study offer some key insights on the mental health needs of LGBTQ+ college students and how public health emergency management can better respond to such needs.

## 5. Conclusions

The survey research assessed the mental health statuses, attitudes towards disease control and mitigation measures, and coping strategies among LGBTQ+ college students due to the COVID-19 pandemic. Our data analysis showed that LGBTQ+ students exhibited higher levels of anxiety and fear compared with the non-LGBTQ+ groups. In addition, a large proportion of LGBTQ+ students were negatively impacted by the pandemic as compared with non-LGBTQ+ groups; they generally held more positive views on the public health measures to alleviate the adverse impacts from COVID-19. It is advisable that public health emergency management should adopt appropriate strategies and adapt their services to support mental health needs of LGBTQ+ students. Our research underscored the need to design tailored health promotion program and to enhance the support systems for LGBTQ+ college students during similar emergencies.

## Figures and Tables

**Table 1 healthcare-12-02047-t001:** Demographics of the Survey Respondents.

Variable		Frequency	%
Sex	Male	124	20
	Female	467	76
	Other	20	3
Sex Orientation	Heterosexual	495	82
	Asexual	24	4
	Bisexual	44	7
	Gay	12	2
	Lesbian	6	1
	Pansexual	10	2
	Queer	9	1
	Unknown	5	2
Race	White	372	61
	Black	118	19
	Latino	47	8
	Asian	53	9
	Native American	16	3
	Other	5	1
Academic Level	Freshman	46	8
	Sophomore	77	13
	Junior	172	28
	Senior	268	44
	Graduate	48	8
Academic Programs	Non-Health-Related	332	54
	Health-Related	279	46
Age	23 and under	566	79
	24–29	28	18
	30–39	12	2
	40 and up	5	1
	Total	611	100

**Table 2 healthcare-12-02047-t002:** *T*-tests for Group Comparisons between LGBTQ+ Students and Non-LGTQ+ Students.

Variables	Non-LGBTQ	LGBTQ	Difference	S.E.	*p*-Value
Anxiety Factors	9.2	11.6	−2.400	0.619	0.0001 **
Fear Factors	−3.65	−1.991	−1.659	0.563	0.0017 **
Trust in Public Health Measures	2.4	4	−1.556	0.443	0.0002 **
Views on Vaccine Safety and Effectiveness	1.2	2.6	−1.300	0.250	0.0000 **
Unhealthy Coping Strategies	−1.18	0.19	−1.369	0.783	0.0404 *
Healthy Coping Strategies	0.931	0.216	0.716	0.457	0.059
Positive Support System	0.141	−0.164	0.305	0.209	0.0727
Negative Support System	−0.653	0.388	−1.040	0.323	0.0007 **

N = 611, * Significant at 0.05, ** Significant at 0.01.

**Table 3 healthcare-12-02047-t003:** Odds Ratios from Logistic Regression for Comparing statuses between LGBTQ+ Students and Non-LGTQ+ Students.

Variables	Non-LGBTQ	LGBTQ	Odds Ratio/Difference	S.E.	*p*-Value
Overall Anxiety	72%	81%	1.652	0.425	0.051
Overall Fear	45%	60%	1.841	0.387	0.004 **
Willingness to Vaccinate	62%	70%	1.439	0.323	0.104
Financial Status Worsened	44%	47%	1.165	0.241	0.461

N = 611, * Significant at 0.05, ** Significant at 0.01.

## Data Availability

The results from this manuscript were generated from an original research project led by the corresponding author, Lei Xu. Currently, there are still remaining findings from this dataset that are under investigation, therefore, they are not publicly available. However, the access to the entire dataset will be available upon request.

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
