# Peer review of "An Assessment of Mental Health Challenges of Lesbian, Gay, Bisexual, and Transgender College Students during the COVID-19 Pandemic"

_healthcare, 2024, doi:10.3390/healthcare12202047_

Round 1

Reviewer 1 Report

Comments and Suggestions for Authors

Thank you for the opportunity to review this manuscript.

 The abstract presents the content of the manuscript well.

The keywords are also appropriate.

The introduction is also appropriate, pertinent and up-to-date, although a little brief.

The methods are presented very briefly. As LGBTQ+ people are a more vulnerable population, they don't mention the type of questions contained in the form, how they guaranteed the anonymity of the participants and the confidentiality of the data. They don't mention whether the study was submitted to and approved by an Ethics Committee. They don't mention the inclusion and exclusion criteria. The description of the statistical part is also very brief and incipient.

The results are well presented, albeit briefly.

The discussion repeats some of the information presented in the results and presents considerations on the subject. It is not very consistent in relation to the presentation of other similar studies with the same population, for possible comparison. It should be improved.

The bibliography used is current and relevant, although 35% of the bibliography used predates the COVID-19 period.

Reviewer 2 Report

Comments and Suggestions for Authors

First of all thanks for your work.

The paper it's well writen, easy to understand and well supported with strong references.

The methods are solid and well chosen.

I just have some suggestion of improvement:

It's interesting if you give more detail about the questionnaire made, number of questions, how much questions about each theme, and if you use some previous used scale or inventory. You evaluate anxiety and fear levels without a scale or inventory for example? Because in results some data is directed for the questions.

You should refer in conclusion that your data show, first of all, a lot of concerns related to mental health problem in college students generally, and specially in LGBTQ+ population.

Congratulations for your work.

Greetings

Reviewer 3 Report

Comments and Suggestions for Authors

An Assessment of Mental Health Challenges of Lesbian, Gay, 2 Bisexual, and Transgender College Students during the COVID-19 Pandemic

Abtract

- The introduction section should provide some background information before presenting the objective.

Method Kindly modify the writing style in this part. In its current form, this is a brief description of the method, lacking some details.

Therefore, it is necessary to revise the content to include additional information. The writing should be categorized into topics, such as

- setting and participants: How can you obtain student email addresses? Who is permitted?  

- questionaire: For example, what is the process for create and develop a questionnaire? What is the number of its parts? What are the components of each part? What are its validity and reliability?

- Data collection: How long do you wait for the questionnaire to be considered a non-response after it has been sent? 

- Ststistic analysis: Which statistics are utilized for each part of data?

Result: The written results must be rewritten in their entirety.

- The presentation should be organized into various parts, including demographic data, mental health, and any additional topics necessary to achieve the objective.  The current content in the results section lacks clarity and is difficult to understand.

- In presenting results, it is advisable to avoid beginning with the term "table." Instead, introduce each topic followed by the results that correspond to it. The table is only an assemblage, not the primary component.

- Each table should be renamed to correspond with its presentation.

- What is the significance level of <0.01 or <0.05? Please select one and provide a rationale in the statistical analysis section of the method section.

- The numerical values should correspond accurately to those presented in the table, rather than relying on estimations.

- There should be three decimal places in the decimal number in the p value portion of the table.

Discussion: The written discussion must be rewritten in their entirety.

- The discussion should be rewritten to correspond with the study's findings and the character of the writing discussion in the manuscript. By addressing significant issues, including comparisons with other studies and the rationale for their similarities or differences with other research.

Conclusion

- Limitations should be move to the discussion section.

Round 2

Reviewer 3 Report

Comments and Suggestions for Authors

An Assessment of Mental Health Challenges of Lesbian, Gay, 2 Bisexual, and Transgender College Students during the COVID-19 Pandemic

Abtract

- The introduction section should provide some background information prior to presenting the objective.

Method 

- setting and participants: The study focuses on college students; however, it states that Registered students who are > 18 years old were recruited, both graduate and undergraduate students are eligible to participate(lines 71-72), raising questions about whether the graduate group qualifies as college students. Please specify whether you are referring to students, a group of master's degree holders, or doctoral degree holders to ensure clarity and avoid any confusion.

Author Response

Please see the attached response.
